# The Impact of Higher Protein Intake in Patients with Prolonged Mechanical Ventilation

**DOI:** 10.3390/nu14204395

**Published:** 2022-10-20

**Authors:** Shih-Wei Huang, Horng-Chyuan Lin, Yu-Feng Chou, Ting-Yu Lin, Chun-Yu Lo, Hung-Yu Huang, Yueh-Fu Fang, Meng-Heng Hsieh, Shu-Min Lin, Yu-Lun Lo, Meng-Jer Hsieh, Kuo-Chin Kao, Chun-Yu Lin, Chung-Chi Huang

**Affiliations:** 1Department of Internal Medicine, Chang Gung Memorial Hospital at Linkou, Taoyuan 333, Taiwan; 2Department of Thoracic Medicine, Chang Gung Memorial Hospital at Linkou, Taoyuan 333, Taiwan; 3College of Medicine, Chang Gung University, Taoyuan 333, Taiwan; 4Department of Nutrition, Chang Gung Memorial Hospital at Linkou, Taoyuan 333, Taiwan

**Keywords:** protein, patients with PMV, prolonged mechanical ventilation, calories, albumin, whey protein

## Abstract

Prolonged mechanical ventilation (PMV) is associated with poor outcomes and a high economic cost. The association between protein intake and PMV has rarely been investigated in previous studies. This study aimed to investigate the impact of protein intake on weaning from mechanical ventilation. Patients with the PMV (mechanical ventilation ≥6 h/day for ≥21 days) at our hospital between December 2020 and April 2022 were included in this study. Demographic data, nutrition records, laboratory data, weaning conditions, and survival data were retrieved from the patient’s electronic medical records. A total of 172 patients were eligible for analysis. The patients were divided into two groups: weaning success (*n* = 109) and weaning failure (*n* = 63). Patients with daily protein intake greater than 1.2 g/kg/day had significant shorter median days of ventilator use than those with less daily protein intake (36.5 vs. 114 days, respectively, *p* < 0.0001). Daily protein intake ≥1.065 g/kg/day (odds ratio: 4.97, *p* = 0.033), daily protein intake ≥1.2 g/kg/day (odds ratio: 89.07, *p* = 0.001), improvement of serum albumin (odds ratio: 3.68, *p* = 0.027), and BMI (odds ratio: 1.235, *p* = 0.014) were independent predictor for successful weaning. The serum creatinine level in the 4th week remained similar in patients with daily protein intake either >1.065 g/kg/day or >1.2 g/kg/day (*p* = 0.5219 and *p* = 0.7796, respectively). Higher protein intake may have benefits in weaning in patients with PMV and had no negative impact on renal function.

## 1. Introduction

For patients with acute respiratory failure, intubation with mechanical ventilation is usually life-saving. However, prolonged mechanical ventilation (PMV) is associated with poorer outcomes and higher economic costs [1,2,3,4,5]. Weaning from the ventilator should be started as soon as possible, and several parameters can be used to predict weaning outcomes in short-term mechanically ventilated patients [6]. The rapid shallow breathing index is the most frequently studied and seems to be an important measuring tool in deciding whether to wean/extubate a patient [7]. However, there have been few studies discussing the predictors of weaning from prolonged mechanical ventilation. In patients with PMV, female sex, body mass index (BMI) > 30 kg/m^2^, hypercapnia, and increased tidal volume/ideal body weight have been proposed as predictors of weaning failure [6,8].

Critical illness is a hypercatabolic state and in the absence of adequate nutrition interventions can predispose patients to malnutrition, leading to poorer clinical outcomes [9]. Interactions between acute metabolic changes, inflammation, and nutrition in early critical illness are complex [10]. Appropriate timing of nutrition therapy and optimal dosing have been suggested, as critical illness and recovery metabolism changes throughout a patient’s course, and energy expenditure and nitrogen losses appear to vary over time [10]. Nutritional therapy strategies for critically ill patients remain controversial. Reignier et al. conducted a randomized, controlled, multicenter study showing that early isocaloric enteral nutrition did not reduce mortality or the risk of secondary infections but was associated with a greater risk of digestive complications in critically ill adults with shock [11]. Weijs et al. found that high, early protein intake (defined as intake at day 4) was associated with lower hospital mortality, while early energy overfeeding was correlated with a higher mortality [12]. Koekkoek et al. proposed that overall, low protein intake was associated with the highest mortality risk. Moreover, the optimal timing of high protein intake may be relevant in intensive care unit (ICU) patients [13].

Nutrition treatment for patients in the early stages of critical illness has been previously studied [11,12,13], but there have been fewer studies that evaluated nutrition and protein intake subscription for patients with PMV. Our study aimed to investigate the effect of nutrition and protein intake on weaning from mechanical ventilation and survival in patients receiving PMV.

## 2. Materials and Methods

This study was a single-center, retrospective analysis of patients with PMV who were admitted to a 24-bed respiratory care center (RCC) at the Linkou Chang Gung Memorial Hospital in Taiwan between December 2020 and April 2022. RCC is a unit that accepted the patient with mechanical ventilation ≥18 days with relative stable clinical condition (hemodynamic stability, free of acute decompensated organ damage and sedative agents) but could not have successful weaning from the ventilator in the ICU for more than 21 days. Patients in RCC receive medical treatment of their disease, underlying causes of respiratory failure, pulmonary rehabilitation, and ventilator weaning. There was a standardized weaning protocol in RCC, and the detailed steps of the protocol were described in previous publications [14,15]. Tracheotomy was recommended if the patients’ state was dependent on ventilator after 14 days of MV. The timing of tracheotomy depended on the decision of doctors and family. The tracheostomy rate tends to be low for cultural reason in Taiwan (lack of acceptance in current Taiwanese society).

The inclusion criteria were as follows: (1) age ≥ 18 years old, (2) PMV (mechanical ventilation ≥6 h/day for ≥21 days), (3) with endotracheal tube or tracheostomy, (4) Acute Physiology and Chronic Health Evaluation II score (APACHE II score) > 10 points, (5) hemodynamic stability without the use of vasopressors or inotropic agents when admitted to the RCC, and (6) without the use of intravenous sedative agent.

The exclusion criteria were as follows: (1) uncontrolled sepsis with unstable hemodynamic status when admitted to the RCC, (2) acute decompensated liver failure or acute decompensated heart failure within 7 days after being admitted to the RCC, (3) mortality within 7 days of being admitted to the RCC, and (4) transferred back to the ICU within 7 days of being admitted to the RCC.

## 3. Data Collection

Demographic and clinical variables were collected from the electronic medical records of our hospital. The collected demographic characteristics were as follows: age, sex, height, weight, ideal body weight, BMI, APACHE II score, length of hospital stay, length of ICU stay (the time spent in ICU before admission to RCC), duration of mechanical ventilation, comorbidity data, weaning results, and mortality status.

## 4. Nutrition

We recorded nutritional data including daily protein intake, daily caloric intake, and the dietary formula for each patient during training for weaning from mechanical ventilation. All the patients received enteral nutrition with formula milk through nasogastric tube. EN involved intermittent bag feeding with 5 to 6 meals per day. The total daily energy requirements were estimated by a dietitian using simplistic formulas (25–30 kcal·kg/day) according to ESPEN and ASPEN guidelines [16]. Protein requirement was calculated as 0.6–1.5 g/kg based on patient condition and underlying diseases. Enteral feeding would be suspended if there were contraindications, such as intestinal obstruction, active gastrointestinal bleeding, or hemodynamic instability. We collected the actual feeding amount according to medical records weekly and analyzed the daily nutrition intake accordingly. Daily whey protein intake was calculated according to the dietary formula. The daily protein intake and daily whey protein intake were measured as grams per kilogram per day (g/kg/day); the daily caloric intake was measured as kilocalories per kilogram per day (kcal/kg/day) according to the actual body weight. We analyzed the best cut-off value for protein intake for weaning and also analyzed the effect of a daily protein intake >1.2 g/kg/day in weaning according to a previous study [12,13].

In the fourth week, we calculated the difference from baseline for serum albumin levels (prescribed as Δ albumin). To evaluate the possible impact of a relatively high daily protein intake on renal function, we collected the serum creatinine level at baseline, second week, and fourth week.

## 5. Outcomes

The primary outcome of this study aimed to evaluate the implications of daily protein intake and nutrition status for weaning. The secondary outcomes included in hospital mortality, the difference of serum albumin level, and serum creatinine level. Successful weaning was defined as a patient who breathed without mechanical ventilator support for 5 consecutive days. Successful weaning and in-hospital mortality were ascertained as of April 2022.

## 6. Ethical Approval

This study was approved by the Institutional Review Board (IRB) of Chang Gung Medical Foundation (approval no.: 202200818B0). The IRB waived the need for the participants’ consent because this was a retrospective study. All personal information was encrypted in a database and anonymized, so there was no breach of privacy.

## 7. Statistical Analysis

Categorical variables were expressed as count and percentage. Parametric data were expressed as mean ± standard deviation (SD). A Student’s *t*-test was used to compare parametric data. Fisher’s exact test was used to compare non-parametric data. We used a receiver operating characteristic (ROC) curve to identify the optimal cut-off-value for the maximum multiplication of sensitivity and specificity. Logistic regression was used for multivariate analysis. Kaplan–Meier survival analysis was performed to assess weaning and survival outcomes. Statistical significance was set at a *p*-value < 0.05. Statistical analyses were performed using GraphPad Prism version 8 (GraphPad Software, La Jolla, CA, USA) and IBM SPSS Statistics 26 (SPSS, Chicago, IL, USA).

## 8. Results

A total of 237 patients with PMV were admitted to our RCC between December 2020 and April 2022, and of these, 65 patients were excluded (Figure 1). In the remaining 172 patients, 109 had weaning success, and 63 had weaning failure. The characteristics of the two groups are summarized in Table 1. There was no significant difference in age, sex, and APACHE II score between the two groups. The BMI and the rate of tracheostomy were higher, and the ICU length of stay was longer in the weaning failure patients. The daily protein intake, daily caloric intake, ∆ albumin, and daily whey protein intake were significantly higher in the successfully weaned patients.

## 9. The Optimal Cut-Off-Value for Nutrition Parameters

The ROC curve and area under the curve (AUC) of daily protein intake, daily caloric intake, ∆ albumin, and daily whey protein intake for a successful weaning outcome are shown in Figure 2. The daily protein intake demonstrated the best predicted value in weaning (AUC: 0.9) and yielded a sensitivity and specificity of 75.2% and 85.7%, respectively, followed by daily caloric intake, ∆ albumin, and daily whey protein intake.

## 10. Prognostic Factors of Weaning Success and in-Hospital Mortality

In the univariate analysis, daily protein intake, daily caloric intake, ∆ albumin, daily whey protein intake, BMI, shorter ICU length of stay, and length of mechanical ventilation before RCC admission had a significant correlation with successful weaning (Table 2). Daily protein intake ≥1.065 g/kg/day (odds ratio: 4.97, *p* = 0.033), daily protein intake ≥1.2 g/kg/day (odds ratio: 89.07, *p* = 0.001), ∆ albumin (odds ratio: 3.68, *p* = 0.027), and BMI (odds ratio: 1.235, *p* = 0.014) had independent significant correlation with successful weaning after multivariate regression analysis (Table 2). Patients with daily protein intake ≥1.065 g/kg/day had significant shorter median days of ventilator use than those with daily protein intake <1.065 g/kg/day (38 vs. 114 days, respectively, *p* < 0.0001). A similar finding was noted among patients with a daily protein intake <1.2 g/kg/day and ≥1.2 g/kg/day (114 days vs. 36.5 days, respectively, *p* < 0.0001). In patients with ∆ albumin < 0.155 g/dL and ≥0.155 g/dL, the median days of ventilator use were 85 days and 53 days, respectively (*p* = 0.0360, Figure 3).

Daily protein intake, daily caloric intake, ∆ albumin, and daily whey protein intake had no correlation with survival (*p* = 0.3155, *p* = 0.7411, *p* = 0.3830, *p* = 0.4118, *p* = 0.5673, respectively, Figure 4, Table 3). BMI and tracheostomy were negatively correlated to mortality (Table 3).

## 11. Renal Function

We analyzed the serum creatinine level at baseline, 2 weeks, and 4 weeks for patients with a daily protein intake ≥1.065 and ≥1.2 g/kg/day separately (Figure 5). In both groups, the serum creatinine level was similar between baseline and the 4th week (*p* = 0.5219 and *p* = 0.7796, respectively). Moreover, the 2nd week serum creatinine level was significantly lower than at baseline (*p* = 0.0294 and *p* = 0.0351, respectively).

## 12. Discussion

To the best of our knowledge, this is the first study to evaluate the clinical implications of higher protein intake in patients with PMV. In this study, the patients with PMV who had a higher protein intake, especially a higher whey protein intake, had better serum albumin improvement, demonstrating a better weaning rate. Moreover, renal function was stable after one month of higher protein intake.

The Society of Critical Care Medicine and The American Society for Parenteral and Enteral Nutrition (ASPEN) published guidelines in 2016 for the “provision and assessment of nutrition support therapy in the adult critically ill patient” [17]. Evidence for protein supplementation was scarce and controversial [17]. However, recently, there has been an increase in evidence focusing on protein supplementation in critically ill patients. In a prospective observational cohort study of 113 ICU patients, Allingstrup et al. found that the higher protein intake group had lower 28-day mortality (low: 0.79 g/kg/day, 27% ICU mortality; medium: 1.06 g/kg/day, 24% ICU mortality; high: 1.46 g/kg/day, 16% ICU mortality) [18]. Nicole et al. demonstrated that 60-day mortality decreased in patients who achieved ≥80% of their prescribed protein intake (target protein intake: 1.2 g/kg/day) [19]. Zusman et al. demonstrated that in critically ill patients with an ICU stay >96 h, administered daily protein >1.3 g/kg/day, resulted in a lower 60-day mortality [20]. Song et al. showed that mechanically ventilated critically ill patients who achieved >90% of their target protein intake (minimal target: 1.2 g/kg/day) had better ICU outcomes, including ICU mortality, in-hospital mortality, weaning from ventilation rate, and ventilation-free days [21]. The European Society for Clinical Nutrition and Metabolism (ESPEN) published new guidelines in 2019 on clinical nutrition in the ICU [22]. The consensus was that the equivalent of 1.3 g/kg protein per day can be delivered progressively during critical illness [22]. In a recently published prospective multinational cohort study in critically ill patients with an ICU stay ≥5 days, a moderate daily macronutrient intake of 10–20 kcal/kg and a protein intake of 0.8–1.2 g/kg were associated with earlier weaning from invasive mechanical ventilation [23].

Even though protein administration for critical ill patients has been investigated for years, those with subacute critical illness with PMV were not defined in either the ASPEN or ESPEN guidelines [17,22]. Among critically ill patients, muscle wasting occurs early and rapidly [24]. Diaphragm dysfunction is present in a high percentage of critically ill patients and is thought to develop due to disuse, secondary to ventilator-induced diaphragm inactivity, and as a consequence of the effects of systemic inflammation, including sepsis [25]. This condition may lead to sustained respiratory failure, increased morbidity, and mortality [25]. During the subacute ICU phase, higher protein/caloric targets should be provided, preferably combined with exercise [10]. In our study, daily protein intake ≥1.065 g/kg/day was correlated with better weaning outcomes. Furthermore, daily protein intake ≥1.2 g/kg/day was an independent predictive factor for successful weaning from mechanical ventilation. Although relatively high protein intake had no impact on the in-hospital mortality rate in our study, we provided a reference for target protein administration during the training period for weaning from mechanical ventilation in patients with PMV.

The 2016 ASPEN guidelines suggested that patients who were at high nutrition risk and accepted >80% of their estimated or calculated target energy and protein within 48–72 h had significant reductions in mortality [17]. For low-risk patients, no correlation was seen between the percentage of target energy delivered and mortality [17]. Braunschweig et al. conducted a prospective randomized trial, which showed that patients with acute respiratory distress syndrome who accepted intensive medical nutrition therapy (provision of >75% of estimated energy and protein needs per day initiated within 6 h of hemodynamic stability) had significantly greater hospital mortality than those who accepted standard nutrition support care (40% vs. 16%, *p* = 0.02) [26]. Other studies also revealed similar findings that early normo-caloric feeding was associated with increased mortality, length of hospital stay, and length of ventilation compared with permissive hypocaloric feeding in critically ill patients [20,27,28,29,30,31]. The 2019 ESPEN guidelines recommended that hypocaloric nutrition (not exceeding 70% of energy expenditure) should be administered in the early phase of acute illness and that after day 3, caloric delivery can be increased up to 80–100% [22]. The negative results of early feeding may contribute to overfeeding with deleterious effects, such as increased length of hospital stay, ventilation duration, infection rates, and the risk of refeeding [10,22]. As for subacute critically ill patients with PMV, optimal calorie administration was not defined in either the ASPEN or ESPEN guidelines [17,22]. In the current study, the administration of calories ≥24.48 kcal/day were also positively correlated to successful weaning. Although the result of multivariate analysis was not significant, there is a trend that sufficient calorie intake may contribute to successful weaning. Target calorie intake with higher protein intake is quite important to patients with PMV.

In 1997, Boirie et al. demonstrated that whey protein and casein had different effects on postprandial protein synthesis [32].Whey protein stimulates protein synthesis but also oxidation; on the other hand, the slowly absorbed casein promotes postprandial protein deposition by inhibiting protein breakdown without an excessive increase in amino acid concentration [32]. Tang et al. investigated the effect of mixed muscle protein synthesis in healthy young men who performed unilateral leg resistance exercises followed by the consumption of either whey, casein, or soy protein [33]. The whey protein stimulated skeletal muscle protein synthesis to a greater extent than either casein or soy, and this effect may be related to the leucine content of the protein consumed and how quickly it is digested [33]. In 2016, Luiking et al. conducted a randomized trial, which showed that whey protein provides a higher rise in serum levels of total amino acids, essential amino acids, and leucine compared with casein in healthy people aged ≥65 years [34]. In the current study, relatively high whey protein intake may lead to better weaning outcomes. This phenomenon may be attributed to its effect on protein synthesis stimulation, enhancing respiratory muscle power.

Brenner et al. demonstrated that dietary protein restriction therapy prevented patients with chronic kidney disease (CKD) from progressive glomerular damage caused by glomerular hypertension [35]. Ko et al. considered that high dietary protein may lead to dilation of afferent arterioles, which results in intraglomerular pressure and glomerular hyperfiltration, and could damage the glomerular structure, particularly in the context of CKD [36]. Nevertheless, in critically ill patients, protein restriction therapy may not be appropriate. In 2021, ESPEN published guidelines on clinical nutrition in hospitalized patients with acute or CKD, which demonstrated that critically ill patients with systemic inflammation and immobilization are strongly catabolic, which induces extensive muscle protein breakdown and impaired protein synthesis [37]. Protein prescription should not be reduced in order to avoid or delay kidney replacement therapy [37]. Our study revealed that the serum creatinine level remained constant after the administration of a relatively high-protein diet in patients with PMV. Short-term increases in protein intake in critically ill patients seems to be a safe strategy for improving weaning outcomes without deteriorating renal function during training for weaning.

## 13. Limitations

There were several inherent limitations in our study. First, this was a retrospective study, and we recorded the dietary prescription and feeding amount weekly. The precise nutritional supplement may have some bias during the weaning period. Second, we did not access or evaluate the nutrition conditions of the patients with successful weaning after they were transferred to the ordinary ward, which may have influenced the in-hospital mortality. Third, we excluded patients who had an APACHE II score < 10, so selection bias may be present. Fourth, nutrition intake was assessed using the actual body weight rather than indirect calorimetry in our study, which may reflect a more accurate nutrition requirement for the patients [22]. Fifth, we did not have the malnutrition assessments before the patients were transferred to RCC, which may have impact on the weaning rate and in hospital mortality.

## 14. Conclusions

In summary, higher daily protein intake may have beneficial effects on weaning from mechanical ventilation in patients with PMV without having negative impact on renal function. The increased whey protein intake and improvement in serum albumin levels were also positive predicting factors for successful weaning. Further prospective, large-scale studies are warranted.

## Figures and Tables

**Figure 1 nutrients-14-04395-f001:**
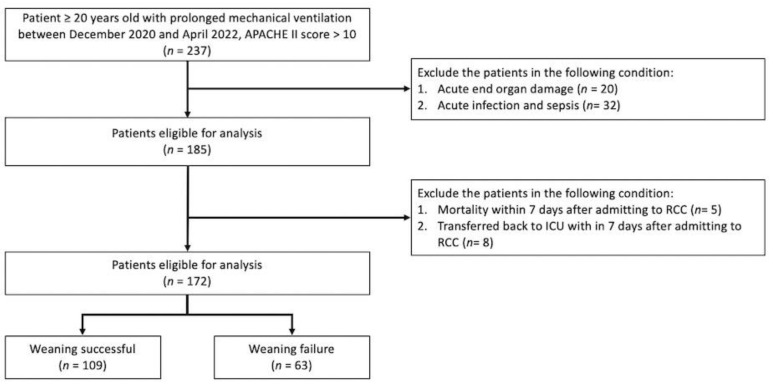
Flowchart of the study design.

**Figure 2 nutrients-14-04395-f002:**
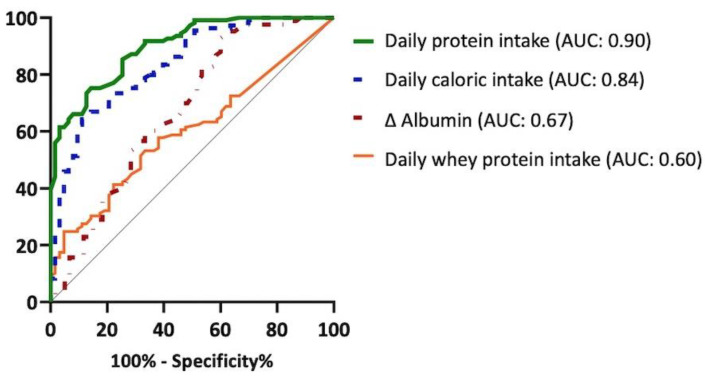
ROC curve of weaning outcomes with nutritional parameters. ROC, receiver operating characteristic; AUC, area under curve; Δ albumin, difference between the 4th week and baseline serum albumin levels.

**Figure 3 nutrients-14-04395-f003:**
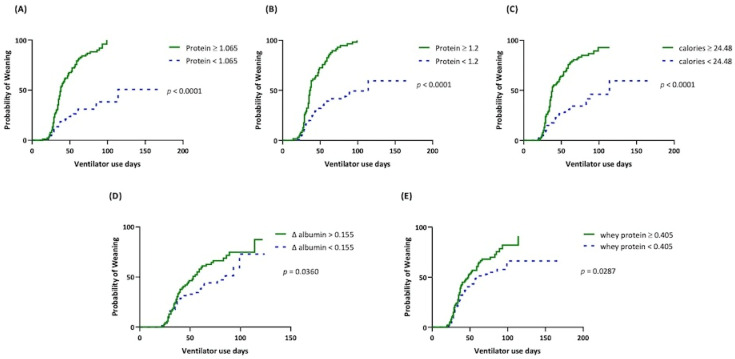
Kaplan–Meier survival curve of weaning outcomes with different nutritional parameters. (**A**) Daily protein intake, g/kg/day; (**B**) daily protein intake, g/kg/day; (**C**) daily calorie intake, kcal/kg/day; (**D**) Δ albumin, difference between the 4th week and baseline serum albumin levels, g/dL; (**E**) daily whey protein intake, g/kg/day.

**Figure 4 nutrients-14-04395-f004:**
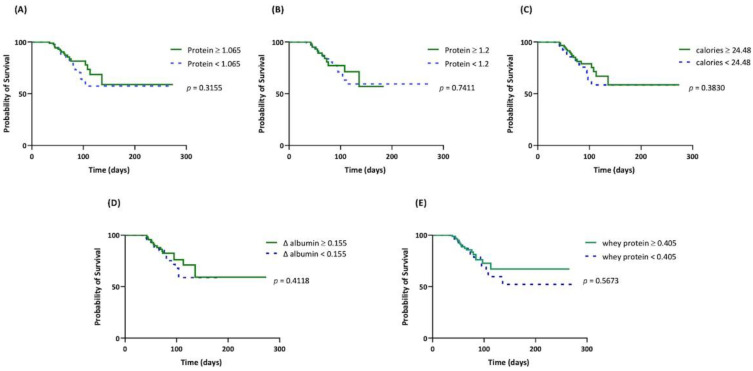
Kaplan–Meier survival curve of survival outcomes for different nutritional parameters. (**A**) Daily protein intake, g/kg/day; (**B**) daily protein intake, g/kg/day; (**C**) daily calorie intake, kcal/kg/day; (**D**) Δ albumin, difference between the 4th week and baseline serum albumin levels, g/dL; (**E**) daily whey protein intake, g/kg/day.

**Figure 5 nutrients-14-04395-f005:**
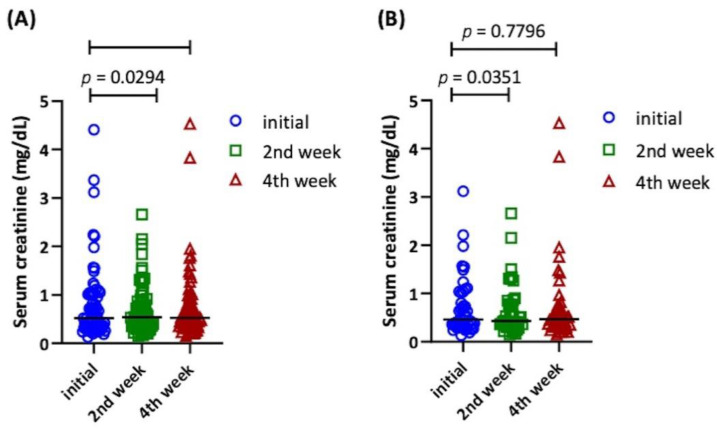
Comparison of baseline, 2nd week, and 4th week serum creatinine levels with different daily protein intakes. (**A**) Daily protein intake ≥1.065 g/kg/day; (**B**) daily protein intake ≥1.2 g/kg/day.

**Table 1 nutrients-14-04395-t001:** Clinical Characteristics of patients with PMV.

Characteristic	Successful, *n* = 109	Fail, *n* = 63	Total, *n* = 172	*p*-Value
Male, n (%)	52 (47.7%)	32 (50.8%)	84 (49%)	0.7525
Age, years, mean ± SD	74.02 ± 12.71	74.63 ± 11.79	74.24 ± 12.35	0.7535
BMI, kg/m^2^, means ± SD	22.96 ± 4.211	24.40 ± 4.485	23.49 ± 4.356	0.0366
APACHE II score, mean ± SD	16.25 ± 4.516	17.65 ± 4.78	16.76 ± 4.65	0.0563
ICU length of stay, mean ± SD	27.63 ± 10.26	31.71 ± 13.14	29.13 ± 11.53	0.0249
ICU indication				
Pneumonia with respiratory failure	53 (48.6%)	32 (50.8%)	83 (%)	0.8744
Septic shock	4 (3.7%)	3 (4.8%)	7 (%)	0.7078
Myocardial infarction	6 (5.5%)	1 (1.6%)	7 (%)	0.4249
Cardiac arrest	7 (6.42%)	9 (14.3%)	16 (%)	0.1052
Acute decompensate heart failure	4 (3.7%)	2 (3.2%)	6 (%)	>0.9999
Intracranial hemorrhage	8 (7.3%)	1 (1.6%)	9 (%)	0.1573
Massive GI bleeding	2 (1.8%)	0 (0%)	2 (%)	0.5333
Ischemic stroke	13 (11.9%)	5 (7.9%)	18 (%)	0.4529
Status epilepsy	5 (4.6%)	2 (3.2%)	7 (%)	>0.9999
Cardiac surgery	1 (0.9%)	4 (6.4%)	5 (%)	0.0608
Abdominal surgery	4 (3.7%)	1 (1.6%)	5 (%)	0.6534
Head and neck surgery	1 (0.92%)	0 (0%)	1 (%)	>0.9999
Thoracic surgery	1 (0.92%)	0 (0%)	1 (%)	>0.9999
Spinal cord injury	0 (0%)	3 (4.8%)	3 (%)	0.0477
Length of mechanical ventilation before RCC admission, mean ± SD	27.18 ± 9.81	31.59 ± 13.11	28.8 ±11.29	0.0133
Tracheostomy, n (%)	36 (33%)	41 (66.1%)	77 (44.8%)	<0.0001
Underlying disease				
Hypertension, n (%)	62 (56.9%)	27 (42.9%)	89 (51.8%)	0.0836
Diabetes mellitus, n (%)	50 (45.9%)	25 (39.7%)	75 (43.6%)	0.5235
Cardiovascular disease, n (%)	54 (49.5%)	32 (50.8%)	86 (50%)	>0.9999
Chronic lung disease, n (%)	18 (16.5%)	13 (20.6%)	31 (18%)	0.5399
Chronic kidney disease, n (%)	35 (32.1%)	20 (31.8%)	55 (32%)	>0.9999
Liver cirrhosis, n (%)	6 (5.5%)	3 (4.8%)	9 (5.2%)	>0.9999
Cerebrovascular disease, n (%)	51 (46.8%)	16 (25.4%)	67 (39%)	0.0060
Malignancy, n (%)	24 (22%)	20 (31.8%)	44 (40.4%)	0.2040
Death, n (%)	20 (18.4%)	19 (30.2%)	39 (22.7%)	0.0898
Death at ward, n (%)	20 (18.4%)	0 (0%)		
Death at RCC, n (%)	0 (0%)	19 (30.2%)		
Nutritional parameters				
Daily protein intake, g/kg/day, mean ± SD	1.29 ± 0.29	0.84 ± 0.23	1.13 ± 0.34	<0.0001
Daily calories intake, kcal/kg/day, mean ± SD	28.04 ± 6.18	19.9 ± 5.83	25.06 ± 7.21	<0.0001
Δ albumin, g/dL, mean ± SD	0.25 ± 0.35	0.002 ± 0.48	0.15 ± 0.43	0.0004
Daily whey protein intake, g/kg/day, mean ± SD	0.58 ± 0.54	0.37 ± 0.36	0.5 ± 0.49	0.0075

BMI, body mass index; APACHE II, Acute Physiology and Chronic Health Evaluation II; Δ albumin, difference between the 4th week and baseline serum albumin levels.

**Table 2 nutrients-14-04395-t002:** Factors associated with weaning outcomes.

Factor	Univariate Regression Analysis			Multivariate Regression Analysis		
	OR	95%CI	*p*-Value	OR	95%CI	*p*-Value
Daily protein intake						
≥1.065 g/kg/day	18.22	7.95–41.74	<0.001	4.97	1.14–21.79	0.033
≥1.2 g/kg/day	75.92	10.16–567.41	<0.001	89.07	6.8–1166.41	0.001
Daily caloric intake						
≥24.48 kcal/kg/day	10.13	4.83–21.25	<0.001	3.78	0.88–16.21	0.073
Δ albumin						
≥0.155 g/dL	2.88	1.44–5.76	0.003	3.68	1.16–11.69	0.027
Daily whey protein intake						
≥0.405 g/kg/day	2.23	1.18–4.20	0.014	2.66	0.82–8.7	0.105
BMI	0.93	0.86–1.00	0.039	1.235	1.04–1.46	0.014
Cerebrovascular disease	2.58	1.31–5.10	0.006	2.43	0.73–8.06	0.146
ICU length of stay	0.97	0.94–1	0.03	1.042	0.81–1.34	0.75
Length of mechanical ventilation before RCC admission	0.97	0.94–0.99	0.018	1	0.77–1.31	0.984
Tracheostomy	0.27	0.14–0.51	0.000	0.08	0.02–0.29	0.000

BMI, body mass index; OR, odds ratio; 95% CI, 95% confidence interval; Δ albumin, difference between the 4th week and baseline serum albumin levels.

**Table 3 nutrients-14-04395-t003:** Factors associated with hospital mortality.

Factor	Univariate Regression Analysis			Multivariate Regression Analysis		
	OR	95%CI	*p*-Value	OR	95%CI	*p*-Value
Daily protein intake						
≥1.065 g/kg/day	0.62	0.3–1.27	0.187	0.87	0.24–3.24	0.839
≥1.2 g/kg/day	0.65	0.3–1.42	0.283	0.27	0.07–1.15	0.077
Daily caloric intake						
≥24.48 kcal/kg/day	0.68	0.33–1.40	0.300	0.69	0.19–2.48	0.573
Δ albumin						
≥0.155 g/dL	0.86	0.40–1.89	0.714	1.08	0.45–2.61	0.857
Daily whey protein intake						
≥0.405 g/kg/day	0.70	0.34–1.43	0.322	0.93	0.39–2.2	0.862
BMI	0.94	0.86–1.02	0.13	0.85	0.74–0.97	0.016
Cerebrovascular disease	1.28	0.62–2.65	0.5	1.42	0.58–3.5	0.444
ICU length of stay	1	0.97–1.03	0.949	1.07	0.9–1.27	0.469
Length of mechanical ventilation before RCC admission	1	0.97–1.03	0.999	0.936	0.78–1.13	0.481
Tracheostomy	0.54	0.26–1.14	0.105	0.34	0.14–0.84	0.019

BMI, body mass index; OR, odds ratio; 95% CI, 95% confidence interval, Δ albumin, difference between the 4th week and baseline serum albumin levels.

## Data Availability

The data presented in this study are available on request from the corresponding author. The data are not publicly available due to local regulations related to medical confidentiality.

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
