# Peer review of "The Impact of Higher Protein Intake in Patients with Prolonged Mechanical Ventilation"

_nutrients, 2022, doi:10.3390/nu14204395_

Round 1

Reviewer 1 Report

This is a very interesting study suggesing that higher protein delivery is correlated with weaning from mechanical ventilation. I had some questions for the authors about some of their results.

1. The odds ratios for ≥1.2 g/kg/day for both univariate (75.92, 95% CI 10.16-567.41, p <0.001) and multivariate regression analysis (89.07, 95% CI 6.8-1166.41, p 0.001) was impressive, but both have tremendously wide confidence intervals. Could the authors comment on why they think the confidence intervals were so wide and what should clinicians take from that? My understanding is that very wide confidence intervals suggest unstable data and less "believable" results.

2. Adequate energy intake also seemed to make a big difference in this patients, and although this is a protein study, it would be important to acknowledge this. Adequate energy delivery is important for protein sparing.

3. The delta albumin is interesting, but is this related to resolving inflammation? Albumin takes weeks-to-months to improve, and it is a poor measure of nutritional status. See Evans DC et al. Nutr Clin Pract 2021;36(1):22-28. Could this change in albumin reflect changes in volume status? Volume overloaded patients may have more difficulty weaning than patients at their dry weight.

4. The significant difference in BMI is interesting; BMI cutoffs for overweight and obesity are lower for Asian populations compared to other populations. Would more of the patients in the failure-to-wean group be considered overweight and do you think this had an impact on weaning?

5. Were there formal malnutrition assessments done for these patients? If so, could this be reported out (eg, if SGA was done, report out SGA A/B/C, etc.)?

Overall, a very interesting study and I think adds signifiantly to the literature that adequate feeding (both energy and protein) is important for weaning from the ventilator.

Author Response

Reviewer 1

This is a very interesting study suggesting that higher protein delivery is correlated with weaning from mechanical ventilation. I had some questions for the authors about some of their results.

  1. The odds ratios for ≥1.2 g/kg/day for both univariate (75.92, 95% CI 10.16-567.41, p <0.001) and multivariate regression analysis (89.07, 95% CI 6.8-1166.41, p <0.001) was impressive, but both have tremendously wide confidence intervals. Could the authors comment on why they think the confidence intervals were so wide and what should clinicians take from that? My understanding is that very wide confidence intervals suggest unstable data and less "believable" results.

Response:

Thanks for the reviewer’s valuable comments.

The confidence intervals will be influenced by the variation in the study groups and the sample size. In our table 1, there were no significant difference between groups (including age, sex, APACHE II score, ICU indication, and underlying disease). The variation in the study groups may have no significant influence on the width of confidence intervals in this study. The wide confidence interval may attribute to small sample size. Further larger prospective studies are warranted to validate our results.

  1. Adequate energy intake also seemed to make a big difference in this patients, and although this is a protein study, it would be important to acknowledge this. Adequate energy delivery is important for protein sparing.

Response:

Thanks for your recommendation.

We had addressed this issue in the discussion section.

“Although the result of multivariate analysis was not significantly, there is a trend that enough calories intake may contribute to successful weaning. Target calorie intake with higher protein intake is quite important to patients with PMV.”

  1. The delta albumin is interesting, but is this related to resolving inflammation? Albumin takes weeks-to-months to improve, and it is a poor measure of nutritional status. See Evans DC et al. Nutr Clin Pract 2021;36(1):22-28. Could this change in albumin reflect changes in volume status? Volume overloaded patients may have more difficulty weaning than patients at their dry weight.

Response:

We appreciate the reviewer’s comments.

We did few analysis for the relationship with volume overloaded, serum albumin level and weaning. Volume overloaded was defined as the patient with documented pulmonary congestion, pulmonary edema or pleural effusion. First, there was no significant difference of volume overloaded between two groups (as the table presented below). Second, in the subgroup analysis of the patients with volume overloaded, the delta albumin level wasn’t correlated the better weaning rate (p value = 0.7667). Therefore, there were no significant correlation of delta albumin with successful weaning and volume status.

  According to the Evans DC et al. Nutr Clin Pract 2021;36(1):22-28 and Soeters, P.B., R.R. Wolfe, and A. Shenkin, JPEN J Parenter Enteral Nutr, 2019. 43(2): p. 181-193, improving serum albumin may be related to resolving inflammation.

Characteristic

Successful,

n = 109

Fail,

n = 63

Total,

n = 172

p value

Volume overload, n (%)

16 (14.7%)

11 (17.5%)

27 (15.7%)

0.6667

  1. The significant difference in BMI is interesting; BMI cutoffs for overweight and obesity are lower for Asian populations compared to other populations. Would more of the patients in the failure-to-wean group be considered overweight and do you think this had an impact on weaning?

Response:

Thanks for your kindly reminder.

In Taiwan, overweighted is defined as BMI 24 kg/m2 ≤ BMI<27 kg/m2 and obesity is defined as the BMI > 27 kg/m2. The correlation of overweighted, obesity and weaning was listed below. None of the overweighted or obesity was correlated with weaning.

Characteristic

Successful,

n = 109

Fail,

n = 63

Total,

n = 172

p value

Overweighted, n (%)

30 (27.5%)

17 (27.0%)

30 (27.3%)

>0.9999

Obesity, n (%)

15 (13.8%)

15 (23.8%)

30 (17.4%)

0.1005

Overweighted and obesity, n (%)

45 (41.3%)

32 (50.8%)

77 (44.8%)

0.2662

  1. Were there formal malnutrition assessments done for these patients? If so, could this be reported out (eg, if SGA was done, report out SGA A/B/C, etc.)?

Response:

Thanks for your kindly reminder.

The formal malnutrition assessments were not performed in this study. We had addressed this issue in the limitation section.

“Fifth, we didn’t have the malnutrition assessments before the patients were transferred to RCC, which may have impact on the weaning rate and in hospital mortality”

Overall, a very interesting study and I think adds signifiantly to the literature that adequate feeding (both energy and protein) is important for weaning from the ventilator.

Reviewer 2 Report

This study investigates the effect of protein intake in days of ventilator use in patients with prolonged mechanical ventilation. The study is a single centre, retrospective study. Studied patients are in weaning after prolonged mechanical ventilation. The study is well performed, and manuscript is clear and concise. Study limitations are well considered.

Some minor changes could improve the manuscript.

1.     1.  Patients were not in an ICU but in a Respiratory Care Centre. It seems that the variable ICU length of stay indicated in table 1 refers to time spent in ICU BEFORE ADMISSION TO Respiratory Care Unit. Please, clarify.

2.  Figure 2 have a low quality and is difficult to interpret. Please, improve it if possible

3.      What was the feeding diet used for the enteral nutrition in patients?

Author Response

Reviewer 2

This study investigates the effect of protein intake in days of ventilator use in patients with prolonged mechanical ventilation. The study is a single center, retrospective study. Studied patients are in weaning after prolonged mechanical ventilation. The study is well performed, and manuscript is clear and concise. Study limitations are well considered.

Some minor changes could improve the manuscript.

  1. Patients were not in an ICU but in a Respiratory Care Centre. It seems that the variable ICU length of stay indicated in table 1 refers to time spent in ICU BEFORE ADMISSION TO Respiratory Care Unit. Please, clarify.

Response:

We appreciate the reviewer’s comments.

We had addressed it in Material and Methods section.

“Demographic and clinical variables were collected from the electronic medical records of our hospital. The collected demographic characteristics were as follows: age, sex, height, weight, ideal body weight, BMI, APACHE II score, length of hospital stay, length of ICU stay (the time spent in ICU before admission to RCC), duration of mechanical ventilation, comorbidity data, weaning results and mortality status.”

  1. Figure 2 have a low quality and is difficult to interpret. Please, improve it if possible

Response:

Thanks for your kindly reminder.

We had modified the quality of Figure 2.

  1. What was the feeding diet used for the enteral nutrition in patients?

Response:

Yes. We had addressed it in Material and Methods section.

“All the patient received enteral nutrition with formula milk through nasogastric tube. EN involved intermittent bag feeding with 5 to 6 meals per day.”

Reviewer 3 Report

Dear Authors,

The manuscript (nutrients-1924328) submitted for review is interesting and if the authors correct it I recommend the manuscript for publication.

 Authors, Please note and address the following comments:

 Abstract

In my opinion, an abstract does not need a structure.

 Keywords: I think, it would be good to use the word "patients with PMV" in keywords.

 Introduction

Authors should use the section Introduction instead of the section Background.

 Material and Methods

This section is well written.

 Results

It is the weakest part of this manuscript.

 References

References are not cited according to journal rules. Publications from MDPI provide information on how to properly cite. Authors may also find this information in the authors' guide.

 The manuscript text uses round brackets incorrectly, they should be square brackets.

 Despite my comments, I am pleased to recommend this manuscript for publication. I believe that it concerns an important area of research in an international context.

 Reviewer

Author Response

Reviewer 3

Dear Authors,

The manuscript (nutrients-1924328) submitted for review is interesting and if the authors correct it I recommend the manuscript for publication.

Authors, Please note and address the following comments:

Abstract

In my opinion, an abstract does not need a structure.

Response:

Thanks for your kindly reminder.

We had corrected.

Keywords: I think, it would be good to use the word "patients with PMV" in keywords.

Response:

Thanks for your recommendation.

We had added in the Keywords.

Introduction

Authors should use the section Introduction instead of the section Background.

Response:

Thanks for your recommendation.

We had modified.

Material and Methods

This section is well written.

Results

It is the weakest part of this manuscript.

 References

References are not cited according to journal rules. Publications from MDPI provide information on how to properly cite. Authors may also find this information in the authors' guide.

 The manuscript text uses round brackets incorrectly, they should be square brackets.

Response:

Thanks for your kindly reminder.

We had corrected.

 Despite my comments, I am pleased to recommend this manuscript for publication. I believe that it concerns an important area of research in an international context.